# The mechanism underlying transient weakness in myotonia congenita

**Jessica H Myers[1], Kirsten Denman[1], Chris DuPont[1], Ahmed A Hawash[2], Kevin R Novak[3], Andrew Koesters[4], Manfred Grabner[5], Anamika Dayal[5], Andrew A Voss[6], Mark M Rich[1]\***

[1]Department of Neuroscience, Cell Biology and Physiology, Wright State University, Dayton, United States; [2]Department of Dermatology & Cutaneous Surgery, University of Miami, Miami, United States; [3]Evokes LLC, Mason, United States; [4]Naval Medical Research Unit, Wright Patterson Air Force Base, Dayton, United States; [5]Department of Pharmacology, Medical University of Innsbruck, Innsbruck, Austria; [6]Department of Biology, Wright State University, Dayton, United States

**Abstract** In addition to the hallmark muscle stiffness, patients with recessive myotonia congenita (Becker disease) experience debilitating bouts of transient weakness that remain poorly understood despite years of study. We performed intracellular recordings from muscle of both genetic and pharmacologic mouse models of Becker disease to identify the mechanism underlying transient weakness. Our recordings reveal transient depolarizations (plateau potentials) of the membrane potential to $-25$ to $-35$ mV in the genetic and pharmacologic models of Becker disease. Both $Na^+$ and $Ca^{2+}$ currents contribute to plateau potentials. $Na^+$ persistent inward current (NaPIC) through $Na_V1.4$ channels is the key trigger of plateau potentials and current through $Ca_V1.1$ $Ca^{2+}$ channels contributes to the duration of the plateau. Inhibiting NaPIC with ranolazine prevents the development of plateau potentials and eliminates transient weakness in vivo. These data suggest that targeting NaPIC may be an effective treatment to prevent transient weakness in myotonia congenita.

**\*For correspondence:**
mark.rich@wright.edu

## Introduction

Myotonia congenita is one of the non-dystrophic muscle channelopathies. It is caused by loss-of-function mutations affecting the muscle chloride channel (ClC-1) (*Lipicky et al., 1971*; *Steinmeyer et al., 1991*; *Koch et al., 1992*). Patients with recessive myotonia congenita (Becker disease) experience muscle stiffness due to hyperexcitability (*Lehmann-Horn et al., 2008*; *Trivedi et al., 2014*; *Cannon, 2015*) as well as transient weakness due to unknown factors (*Ricker et al., 1978*; *Rüdel et al., 1988*; *Zwarts and van Weerden, 1989*; *Deymeer et al., 1998*; *the CINCH Consortium et al., 2013*). Some patients with Becker disease report transient weakness in arm muscles as a greater impediment than muscle stiffness (*Rüdel et al., 1988*). This weakness can last up to 90 s and is brought on by exertion following rest (*Ricker et al., 1978*; *Rüdel et al., 1988*; *Zwarts and van Weerden, 1989*; *Deymeer et al., 1998*).

The mechanism underlying transient weakness in Becker disease has remained unknown since its initial description close to 50 years ago (*Ricker and Meinck, 1972*). There appears to be loss of muscle excitability, as weakness is accompanied by a drop in compound muscle action potential (CMAP) amplitude during repetitive stimulation (*Ricker and Meinck, 1972*; *Brown, 1974*; *Aminoff et al., 1977*; *Deymeer et al., 1998*; *Drost et al., 2001*; *Modoni et al., 2011*). This drop in CMAP is associated with reduction in muscle fiber conduction velocity, which has been proposed to progress to depolarization block (*Zwarts and van Weerden, 1989*). What has remained unclear, and perhaps counterintuitive, is why a loss-of-function mutation of the muscle ClC-1 channels in myotonia

**eLife digest** Myotonia is a neuromuscular condition that causes problems with the relaxation of muscles following voluntary movements. One type of myotonia is Becker disease, also called recessive myotonia congenita. This is a genetic condition that causes muscle stiffness as a result of involuntary muscle activity. Patients may also suffer transient weakness for a few seconds or as long as several minutes after initiating a movement. The cause of these bouts of temporary weakness is still unclear, but there are hints that it could be linked to the muscle losing its excitability, the ability to respond to the stimuli that make it contract. However, this is at odds with findings that show that muscles in Becker disease are hyperexcitable.

Muscle excitability depends on the presence of different concentrations of charged ions (positively charged sodium, calcium and potassium ions and negatively charged chloride ions) inside and outside of each muscle cells. These different concentrations of ions create an electric potential across the cell membrane, also called the 'membrane potential'. When a muscle cell gets stimulated, proteins on the cell membrane known as ion channels open. This allows the flow of ions between the inside and the outside of the cell, which causes an electrical current that triggers muscle contraction.

To better understand the causes behind this muscle weakness, Myers et al. used mice that had either been genetically manipulated or given drugs to mimic Becker disease. By measuring both muscle force and the electrical currents that drive contraction, Myers et al. found that the mechanism underlying post-movement weakness involved a transient change in the concentrations of positively charged ions inside and outside the cells. Further experiments showed that proteins that regulate the passage of both sodium and calcium in and out of the cell – called sodium and calcium channels – contributed to this change in concentration. In addition, Myers et al. discovered that using a drug called ranolazine to stop sodium ions from entering the cell eliminated transient weakness in live mice.

These findings suggest that in Becker disease, muscles cycle rapidly between being hyperexcited or not able to be excited, and that targeting the flow of sodium ions into the cell could be an effective treatment to prevent transient weakness in myotonia congenita. This study paves the way towards the development of new therapies to treat Becker disease as well as other muscle ion channel diseases with transient weakness such as periodic paralysis.

congenita (*Lipicky et al., 1971*; *Steinmeyer et al., 1991*; *Koch et al., 1992*) leads to transient loss of excitability. The primary defect caused by loss of ClC-1 current is hyperexcitability of muscle, which causes myotonia.

We established that a ClC-1 homozygous null (ClC$^{adr}$) mouse model of Becker disease has transient weakness in vivo, mimicking the condition in human patients. Intracellular recording in both ClC$^{adr}$ muscle and a pharmacologic model of Becker disease (due to block of ClC-1 with 9-AC) have elucidated a novel phenomenon: transient depolarizations to voltages between $-25$ and $-35$ mV, lasting many seconds, which we termed 'plateau potentials.' Blocking Na$^+$ persistent inward current (NaPIC) with ranolazine prevented both development of plateau potentials and transient weakness. We conclude that NaPIC plays a central role in the development of plateau potentials, which are the mechanism underlying transient weakness in Becker disease.

## Results

### Identification of transient weakness in mice with myotonia congenita

To study in vivo isometric motor performance in the *Clcn1*$^{adr-mto2J}$ (ClC$^{adr}$) mouse model of recessive myotonia congenita (Becker disease), a muscle force preparation that we used previously was employed (*Dupont et al., 2019*; *Wang et al., 2020*). Mice were anesthetized via isoflurane inhalation and the distal tendon of the triceps surae (gastrocnemius, plantaris, and soleus muscles) was dissected free and attached to a force transduction motor; then the sciatic nerve was stimulated with 45 pulses delivered at 100 Hz. In unaffected littermates, there was no myotonia following 45 pulses at 100 Hz, such that relaxation was immediate (*Figure 1A*). In ClC$^{adr}$ mice, stimulation with 45 pulses caused full fusion of force, but relaxation was slowed, due to the presence of myotonia (*Figure 1C*).

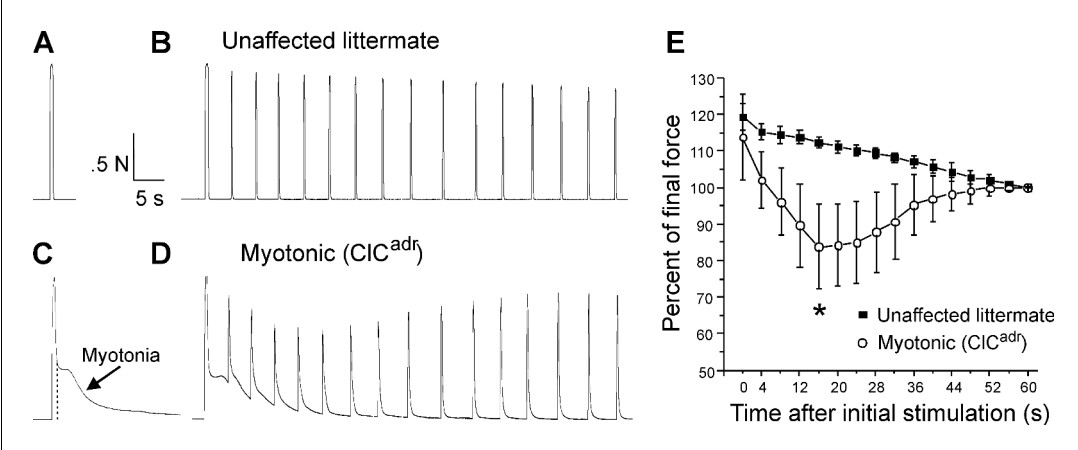

**Figure 1.** Transient weakness in the ClC$^{adr}$ mouse model of recessive myotonia. (**A**) Shown is a force trace from a triceps surae muscle group of an unaffected littermate in response to stimulation of the sciatic nerve with 45 pulses at 100 Hz. (**B**) The force trace generated by following the initial stimulus with 15 pulses at 100 Hz delivered every 4 s. (**C**) In a ClC$^{adr}$ mouse, following the 45 pulses at 100 Hz (indicated by the dotted vertical line) there was continued force generation secondary to myotonia. (**D**) With stimulation every 4 s transient weakness was revealed. (**E**) Plot of the force normalized to force at 60 s in unaffected littermates and ClC$^{adr}$ mice. Transient reduction in force was present in ClC$^{adr}$ mice (*, p=0.00015 vs unaffected littermates at 16 s, t-test, 95% confidence interval 110–114 vs 75–93). n = 5 unaffected littermates and n = 9 ClC$^{adr}$ mice. Error bars represent ± SD.

To determine whether transient weakness was present, the sciatic nerve was additionally stimulated with 15 pulses at 100 Hz every 4 s for 1 min. In unaffected littermates, this caused stable force production with a mild, gradual reduction that was likely due to fatigue (**Figure 1B**). In myotonic mice, the same stimulation protocol revealed transient weakness, as force fell over the first 10–15 s and then recovered (**Figure 1D**). To avoid inclusion of fatigue in measurement of transient weakness, we normalized to force at the end of the 1 min of intermittent stimulation. The plot of the mean normalized force revealed transient weakness, which peaked in severity 15–20 s after the initial stimulation and resolved within 1 min (**Figure 1E**). These data indicate that transient weakness is present in ClC$^{adr}$ mice.

## Characterization of plateau potentials in genetic and pharmacologic models of myotonia congenita

In myotonic patients, transient weakness is paralleled by a drop in CMAP amplitude (*Ricker and Meinck, 1972*; *Modoni et al., 2011*). This finding suggests that weakness is due to loss of excitability. To look for inexcitability of ClC$^{adr}$ muscle, intracellular current clamp recordings were performed. In both unaffected littermates and ClC$^{adr}$ mice, stimulation with a 200 ms injection of depolarizing current triggered repetitive firing of action potentials during the stimulus. In muscle from unaffected littermates, the firing ceased as soon as the stimulus was terminated (**Figure 2A**). In muscle from ClC$^{adr}$ mice, there was myotonia (continued firing of action potentials following termination of the stimulus) in 100% of fibers (**Figure 2B**). The myotonia often persisted for many seconds. While most runs of myotonia ended with repolarization to the resting membrane potential (**Figure 2B**), in some instances, myotonia terminated with the development of depolarizations lasting 5 to >100 s to a membrane potential near −35 mV (**Figure 2C,D**). During these prolonged depolarizations, there was a gradual repolarization of the membrane to near −45 mV, followed by sudden repolarization back to the resting potential. In some cases, the sudden repolarization was preceded by the development of oscillations in the membrane potential (**Figure 2D**). The prolonged depolarizations occurred in 30% of ClC$^{adr}$ muscle fibers (n = 36/119 fibers from 10 mice). Of the 36 fibers with plateau potentials, 26 repolarized to within 4 mV of their initial resting potential. The 10 fibers that did not fully repolarize may have become damaged and thus were not analyzed.

Depolarizations lasting less than 1 s to a membrane potential close to −60 mV have been described in a toxin-induced model of hyperkalemic periodic paralysis and were termed plateau depolarizations (*Cannon and Corey, 1993b*). While the depolarizations we identified could be due

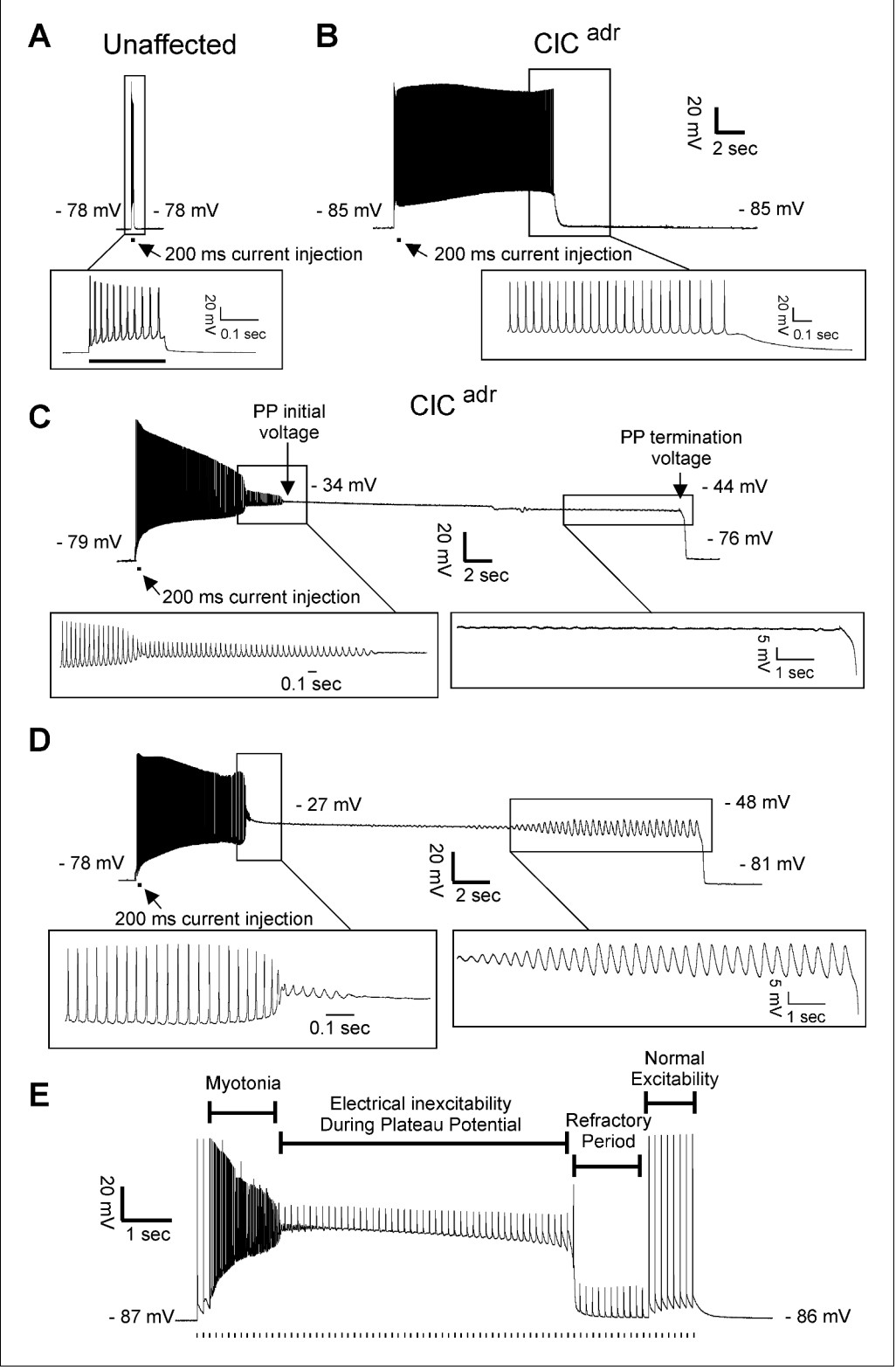

**Figure 2.** Plateau potentials in ClC[adr] muscle. For **A–D**, the insets show portions of the traces on an expanded time base. (**A**) The response of muscle from an unaffected littermate to injection of 200 ms of depolarizing current (horizontal bar below the voltage trace); note that action potentials stop when current stops. (**B–D**) Traces of myotonia triggered by a 200 ms injection of depolarizing current from three different ClC[adr] muscle fibers. The following membrane potentials are identified in **C** and **D**: membrane potential prior to stimulation, initial

*Figure 2 continued on next page*

*Figure 2 continued*

membrane potential during the plateau potential, membrane potential prior to the termination of the plateau potential, and membrane potential following repolarization. (E) Development of a plateau potential during repetitive stimulation at 8 Hz (stimuli represented by vertical hash marks under the recording).

to similar mechanisms as those described in hyperkalemic periodic paralysis, they seemed more similar to prolonged depolarizations in spinal motor neurons, which can last many seconds, and have been termed plateau potentials (*Alaburda et al., 2002*; *Heckman and Enoka, 2012*; *Hounsgaard, 2017*). We thus chose the term plateau potentials to describe them.

To determine whether plateau potentials cause inexcitability, ClC$^{adr}$ muscle fibers were stimulated during the plateau phase. During and immediately following plateau potentials, action potential generation in response to current injection failed (*Figure 2E*). At later times following repolarization, action potential generation was again possible. The inexcitability at earlier times following repolarization is likely due to slow inactivation of Na channels following the many-second depolarization (*Ruff, 1996*; *Ruff, 1999*; *Rich and Pinter, 2003*). These data are consistent with the possibility that plateau potentials and the resultant inexcitability of muscle are the mechanisms underlying transient weakness in myotonia congenita.

In ClC$^{adr}$ muscle, ClC-1 chloride conductance has been absent throughout development such that plateau potentials could be a compensatory response to muscle hyperexcitability. To test this possibility, we acutely blocked ClC-1 chloride channels in muscle from unaffected littermates with 100 µM 9-anthracene carboxylic acid (9-AC). This dose of 9-AC blocks more than 95% of ClC-1 chloride channels in skeletal muscle (*Palade and Barchi, 1977*) and has been used to model myotonia congenita both in vitro and in vivo (*van Lunteren et al., 2011*; *Desaphy et al., 2013*; *Desaphy et al., 2014*; *Skov et al., 2015*). Acute blocking of ClC-1 triggered myotonia and plateau potentials in 92% of fibers (*Figure 3A*, 49/53 fibers from eight mice). Following acute block of ClC-1 channels with 9-AC, all plateau potentials terminated with sudden repolarization to within 4 mV of the previous resting potential (49/49 fibers). These data strongly suggest that the ion channels responsible for development of plateau potentials are present in wild-type skeletal muscle.

In contrast to the relative consistency in voltage at onset and termination of plateau potentials, the duration in both the ClC$^{adr}$ and 9-AC treated models of myotonia was highly variable (*Figure 3B*, *Table 1*, variance = 3131 s for ClC$^{adr}$ and 623 s for 9-AC treated muscles). The reason for the high variance of duration was that the rate of repolarization during the plateau potential varied by more than 100-fold (*Figure 3B*, *Table 1*). It was generally not possible to record multiple plateau potentials in individual ClC$^{adr}$ fibers due to the long median duration. However, it was possible to record multiple plateau potentials within individual fibers of 9-AC treated muscle, as most lasted only a few seconds. Within individual fibers, the slope of repolarization of plateau potentials was less variable such that duration was relatively constant with a mean variance of 0.5 s ± 0.7 s (n = 17 9-AC treated fibers in which four or more plateau potentials were recorded).

In both ClC$^{adr}$ muscle and 9-AC treated muscle, plateau potentials did not occur following every run of myotonia (*Figure 3C*). To determine why some runs of myotonia ended in plateau potentials while others did not, we compared the mean voltage at the end of runs of myotonia that produced plateau potentials vs. the mean voltage at the end of runs that did not produce plateau potentials, in 9-AC treated fibers. There was a strong correlation between the mean membrane potential during the final 500 ms of runs of myotonia and development of plateau potentials. In 22/22 fibers, the mean membrane potential was more depolarized in runs of myotonia that produced plateau potentials (mean = −40.1 ± 3.1 mV vs −49.0 ± 4.5 mV, p=5 × 10$^{-11}$, paired t-test, *Figure 3D*). This suggested that a voltage-dependent current might be involved. However, another feature of myotonia that determines whether a plateau potential is triggered is the firing rate, which might correlate with changes such as build-up of K$^+$ in t-tubules or elevation of intracellular Ca$^{2+}$. Thus, we examined and found a strong correlation between the firing rate of myotonia runs and subsequent development of plateau potentials: In 21/22 fibers, the mean firing rate was higher for runs of myotonia ending in plateau potentials (mean = 31.2 ± 5.7 Hz vs 23.4 ± 2.9, p=7 × 10$^{-6}$, paired t-test, *Figure 3E*). Thus, while voltage-dependent channels appear to be involved, changes in ion concentrations due to differences in firing rates remain a possible contributor.

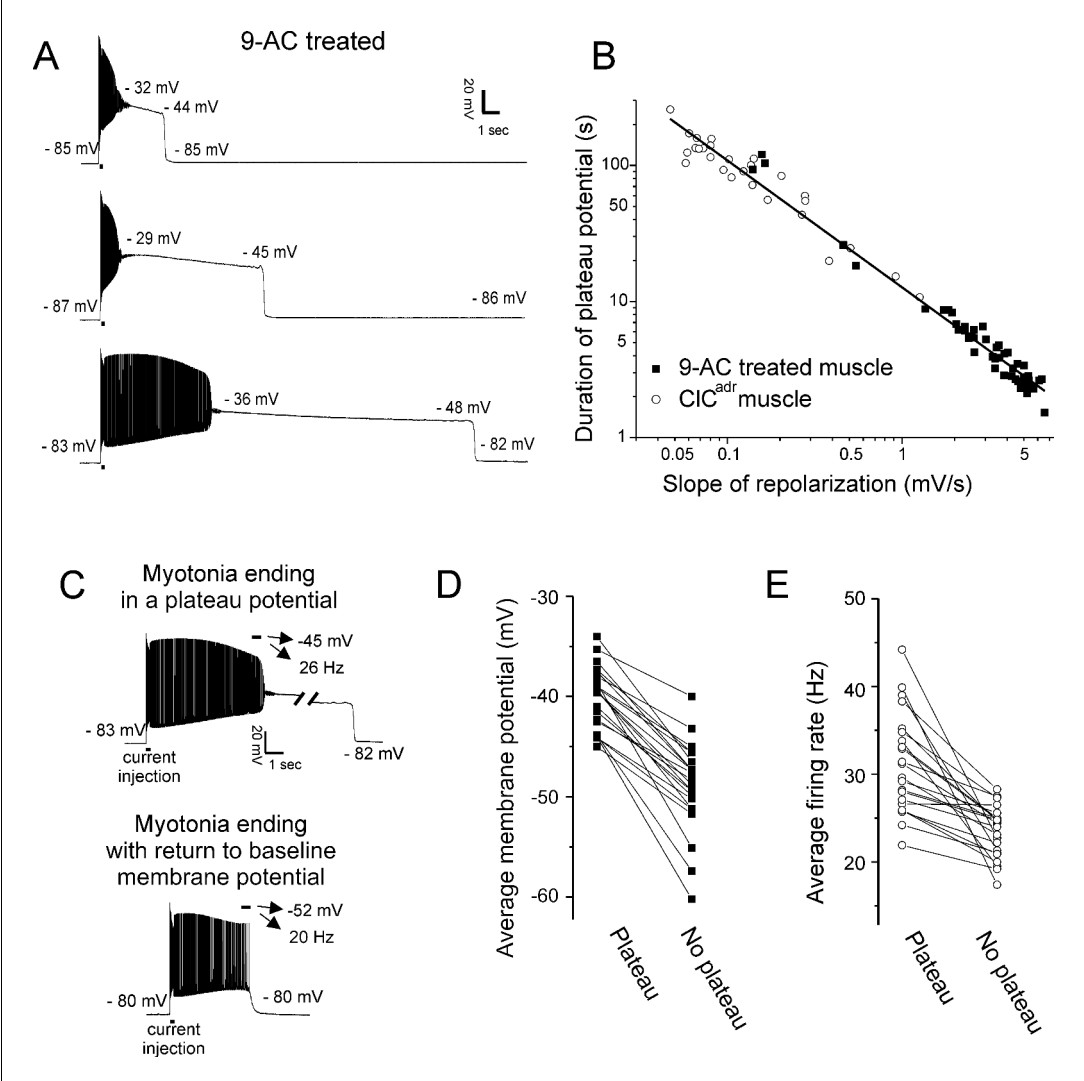

**Figure 3.** Characterization of plateau potentials. (**A**) Three examples of plateau potentials of different duration in 9-AC treated muscle. The 200 ms current injection is indicated by a horizontal bar underneath the trace. Indicated on each trace is the membrane potential at the start and end of the plateau potential. (**B**) The duration of plateau potentials in the ClC[adr] and 9-AC models of myotonia plotted against the rate of repolarization during the plateau potential. A linear fit was performed on the log–log plot with an $R^2$ value of 0.98 and a slope of −0.93. (**C**) Shown are two runs of myotonia from the same muscle fiber, one ending in a plateau potential and one ending with repolarization. The mean membrane potential and firing rate in the final 500 ms of myotonia (horizontal bar) are indicated by the arrows above each trace. (**D**) Plot of the average membrane potential during the final 500 ms of myotonia for 22 fibers in which there were both a run of myotonia ending in a plateau potential and a run ending with repolarization (no plateau potential). (**E**) Plot of the average firing rate during the final 500 ms of myotonia for the same 22 fibers.

Another factor that might determine whether plateau potentials are triggered is the duration of the preceding myotonia. However, recordings of multiple plateau potentials in individual 9-AC treated muscle fibers did not reveal a consistent pattern of duration of myotonia prior to entry into plateau potentials (*Figure 4*). While this does not rule out a contribution of duration of myotonia, it suggests other factors predominate.

**Table 1.** Plateau potential parameters in ClC[adr] and 9-AC treated unaffected muscle.

| | Pre-myotonia vm (mV) | Initial PP vm (mV) | PP repol slope (mV/s) | Final PP vm (mV) | Max repol rate (mV/s) | Post PP vm (mV) | Duration of PP (s) |
|---|---|---|---|---|---|---|---|
| ClC[adr] | −80.4 ± 2.9 (−80.0) | −35.2 ± 4.2 (−35.8) | −0.25 ± 0.19 (−0.17) | −46.3 ± 2.9 (−46.7) | 250 ± 78 (246) | −80.4 ± 2.9 (−79.7) | 81.9 ± 47.6 (74.4) |
| 9-AC | −81.5 ± 2.6 (−82.4) | −31.2 ± 3.1* (−31.4) | −3.4 ± 1.8** (−3.5) | −45.2 ± 2.3 (−45.1) | 197 ± 44[+] (192) | −80.8 ± 2.9 (−81.6) | 11.7 ± 25.0[++] (4.1) |

Shown are the mean value ± the standard deviation for parameters of plateau potentials in ClC[adr] fibers and 9-AC treated fibers. Below the mean value is the median value in parenthesis. n = 26 fibers with plateau potentials from 10 ClC[adr] mice and 49 fibers with plateau potentials from eight unaffected muscles treated with 9-AC. Vm is membrane potential, and repol is repolarization. Pre-myotonia Vm is the resting membrane potential prior to stimulation. Initial PP Vm is the membrane potential at the beginning of the plateau potential. Final PP Vm is the membrane potential prior to the sudden repolarization terminating the plateau potential. Max repol rate is the maximum rate of repolarization during the sudden repolarization. Post PP Vm is the membrane potential following termination of the plateau potential. * indicates p<0.05, ** indicates $p < 1 \times 10^{-20}$ of log transformed data. [+] indicates p<0.01. [++] indicates $p < 1 \times 10^{-18}$ of log transformed data.

## Ca²⁺ and Na⁺ currents contribute to generation of plateau potentials

To determine whether development of plateau potentials is due primarily to opening or closing of ion channels, we measured the membrane response to injection of hyperpolarizing or depolarizing square current pulses at baseline and during plateau potentials. Injection of 5 nA of either depolarizing or hyperpolarizing current during plateau potentials led to variable responses that early in the plateau potential appeared passive, but prior to repolarization triggered an overshoot of membrane potential after termination of current injection (*Figure 5*). The presence of an overshoot fits with the propensity of membrane potential to oscillate prior to termination of plateau potentials (*Figure 2D*). Since the response appeared passive during the early phase of plateau potentials we estimated relative input resistance, which was reduced by 50% following initial depolarization versus baseline (0.43 ± 0.05 vs 0.90 ± 0.12 MΩ, p<0.01, paired t-test, n = 14 fibers). While caution must be used in interpreting these data, they favor the possibility that there is a net increase in membrane conductance during the early phase of plateau potentials.

One possible explanation for the overshoot in voltage following termination of current injection during the late phase of plateau potentials is voltage-dependent opening and closing of $K_V$ channels. The voltage near the end of plateau potentials is near the midpoint of activation of rodent skeletal muscle $K_V$ channels (*Beam and Donaldson, 1983*). With the injection of hyperpolarizing current, $K_V$ channels would be caused to close, resulting in a depolarizing overshoot following termination of current injection. With the injection of depolarizing current, the converse would happen. This may not occur during the initial phase of plateau potentials if $K_V$ channels are mostly open or inactivated (*DiFranco et al., 2012*).

The finding that membrane conductance is increased during the initial phase of plateau potentials suggests opening of either Ca²⁺ or Na⁺ channels. In spinal motor neurons, L-type Ca²⁺ channels play a central role in generation of plateau potentials (*Alaburda et al., 2002*; *Heckman and Enoka, 2012*; *Hounsgaard, 2017*). To determine whether Ca²⁺ current through skeletal muscle L-type channel ($Ca_V1.1$) triggers plateau potentials, we performed recordings on skeletal muscle fibers from a mouse model (*ncDHPR*) in which the pore region of $Ca_V1.1$ carries a point mutation leading to ablation of inward Ca²⁺ current (*Dayal et al., 2017*). We used voltage clamp of FDB/IO fibers to verify the absence of inward currents in *ncDHPR* mice. In wild-type littermate controls, ramp depolarization following block of Na⁺, K⁺, and Cl⁻ channels triggered a large, inward Ca²⁺ current, which began to activate at −15.1 ± 6.9 mV (n = 4 fibers) with a mean amplitude of 165 ± 46 nA (*Figure 6A,B*). Ramp depolarization triggered no inward Ca²⁺ current in *ncDHPR* muscle fibers (mean = 0 ± 0 nA, n = 10 fibers; *Figure 6C*).

To determine whether current flow through $Ca_V1.1$ contributes to generation of plateau potentials, current clamp recordings were performed. Application of 100 μM 9-AC led to the development of plateau potentials in 41/49 fibers from four wild-type mice and in 47/49 fibers from three *ncDHPR* mice (*Figure 6D and E*). Analysis of the characteristics of plateau potentials suggests that Ca²⁺ current flow through $Ca_V1.1$ channels influences the duration of the plateau. While there was no difference in the beginning or ending voltages of plateau potentials, the duration was

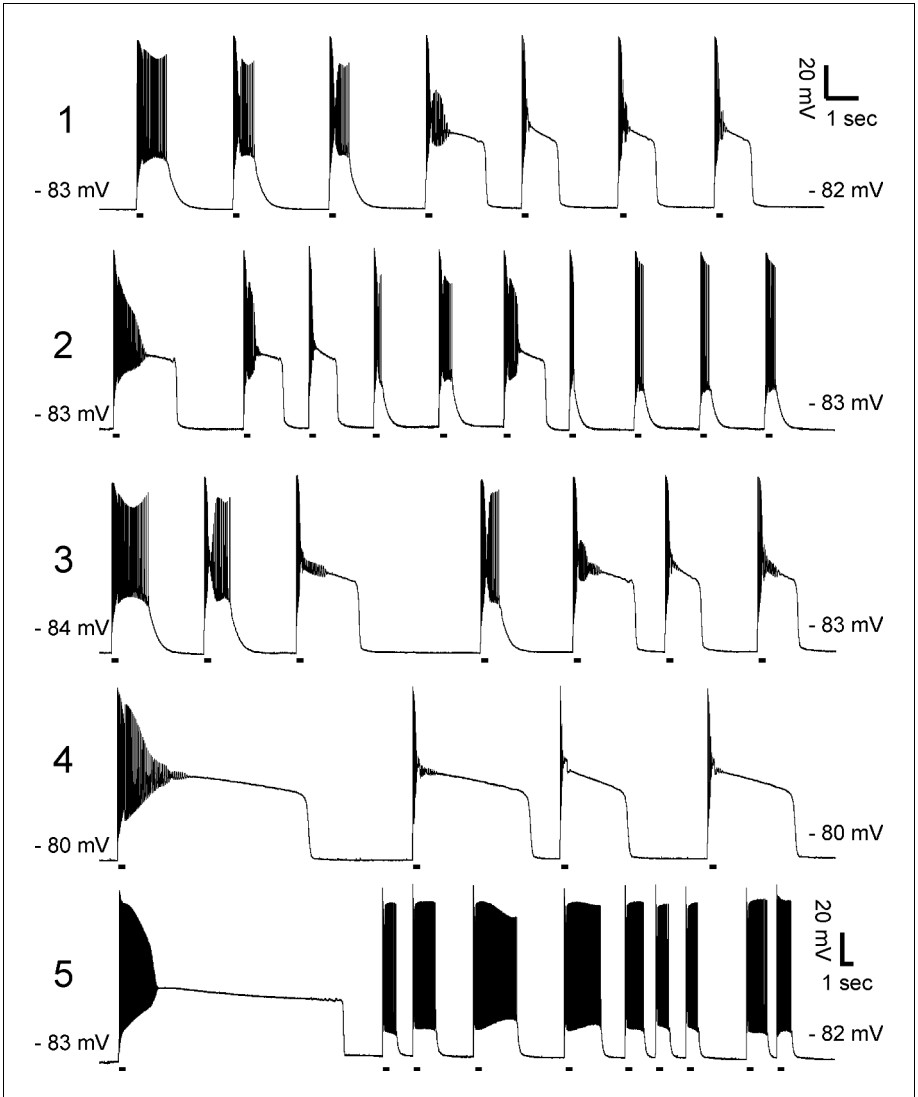

**Figure 4.** Lack of correlation between duration of myotonia and development of plateau potentials. Exemplary prolonged intracellular recordings of plateau potentials from 9-AC treated muscle fibers (n = 5 fibers). There is no consistent relationship between duration of myotonia and development of plateau potentials. The horizontal lines under each trace mark 200 ms injections of depolarizing current.

shorter in *nc*DHPR muscle (*Figure 6D and E*, *Table 2*). The cause of the shorter plateau potential was an increase in the rate of repolarization (*Table 2*). These data suggest that $Ca^{2+}$ influx through $Ca_v1.1$ does not play a role in the initiation of plateau potentials, but is involved in sustaining them.

The $Na_v1.4$-mediated $Na^+$ current responsible for the generation of action potentials in skeletal muscle inactivates within ms of depolarization, making it highly unlikely that it contributes to development of plateau potentials. However, a $Na^+$ current lacking fast inactivation (NaPIC) was found to be involved in the generation of plateau depolarizations in muscle in a toxin model of hyperkalemic periodic paralysis (*Cannon and Corey, 1993b*). We recently determined that NaPIC contributes to repetitive firing occurring during myotonia (*Hawash et al., 2017*; *Metzger et al., 2020*). NaPIC is present in normal skeletal muscle and likely derives from modal gating in which a small subset of $Na_v1.4$ channels reversibly enter a mode lacking fast-inactivation (*Patlak and Ortiz, 1986*; *Gage et al., 1989*). We term the $Na^+$ channels responsible for action potentials 'fast-inactivating $Na^+$ channels' and $Na^+$ channels lacking fast inactivation 'NaPIC'.

To determine whether NaPIC might play a role in the generation of plateau potentials, we applied ranolazine to 9-AC treated muscle. Ranolazine has been found to preferentially block NaPIC

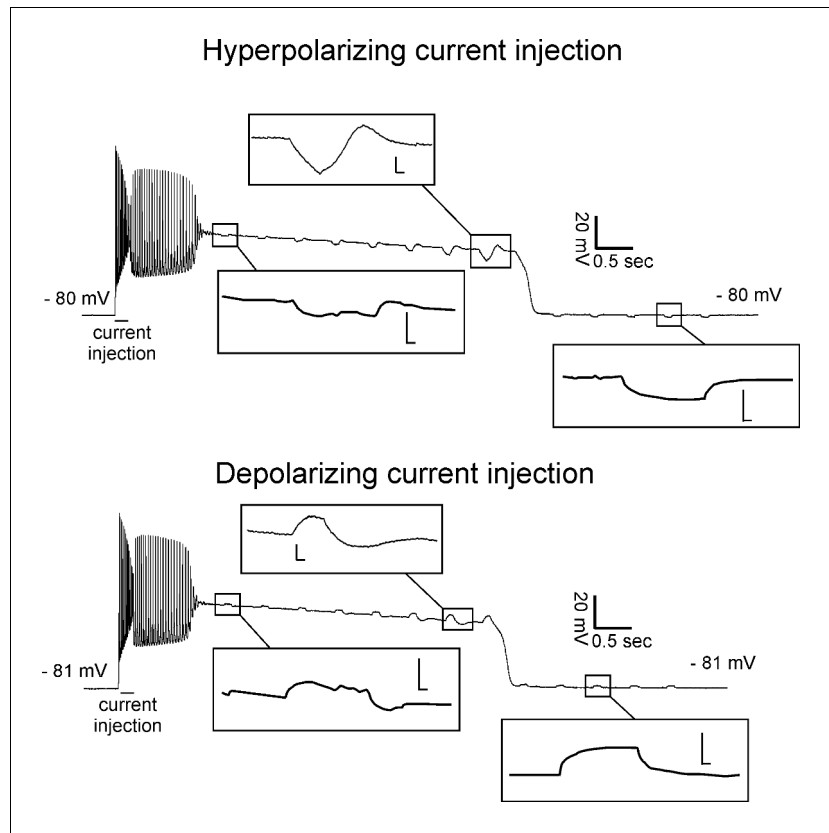

**Figure 5.** Reduced input resistance during the early phase of plateau potentials. Shown are recordings with 200 ms injections of hyperpolarizing (top) or depolarizing (bottom) current during two plateau potentials from a 9-AC treated fiber. Insets show the trace on an expanded voltage and time axis (scale bars of insets 4.0 mV and 0.02 s).

in brain, heart, peripheral nerve, and skeletal muscle (*El-Bizri et al., 2011*; *Kahlig et al., 2014*). We previously found that ranolazine was effective in eliminating myotonia by blocking NaPIC while sparing enough fast-inactivating Na$^+$ channels to allow for repetitive firing of action potentials triggered by current injection (*Novak et al., 2015*; *Hawash et al., 2017*). As shown in *Figure 7A*, when myotonia was triggered by the treatment of muscle with 9-AC, 94% of fibers (n = 53 fibers from eight mice) developed plateau potentials in response to a 200 ms injection of depolarizing current. Following treatment with 40 µM ranolazine, 0/22 fibers from three mice developed plateau potentials after 200 ms current injection (*Figure 7A*, p<0.01 vs untreated). These data were consistent with NaPIC playing a role in the development of plateau potentials.

However, as myotonia was greatly reduced by ranolazine (*Novak et al., 2015*; *Hawash et al., 2017*), it was possible that elimination of plateau potentials was secondary to a reduction in the number of myotonic action potentials. We thus changed our stimulation protocol to a 2 s train of 3 ms stimulus pulses delivered at 20 Hz. This firing rate and duration of firing mimics the duration and rate of firing during runs of myotonia (*Hawash et al., 2017*). For trains of stimuli, the amplitude of 3 ms pulses of current were first adjusted to find the lowest current required to elicit an action potential. The current was then increased by 10 nA prior to delivering a train of stimuli. 2 s of 20 Hz stimulation triggered plateau potentials in 46/55 fibers (n = 5 mice, *Figure 7B*). Out of the 46 fibers with plateau potentials triggered by 20 Hz trains of stimuli, 11 fibers entered plateau potentials with limited (three or less myotonic APs) or no myotonia. When 40 µM ranolazine was applied, plateau potentials developed in 0/68 fibers (n = 5 mice, p<0.01 vs untreated, *Figure 7B*). These data suggest that ranolazine is not eliminating plateau potentials via a secondary effect of prevention of repetitive firing.

As shown in *Figure 3C and D*, the mean membrane potential at the end of a run of myotonia correlated with whether myotonia terminated in a plateau potential or with repolarization. Thus, we

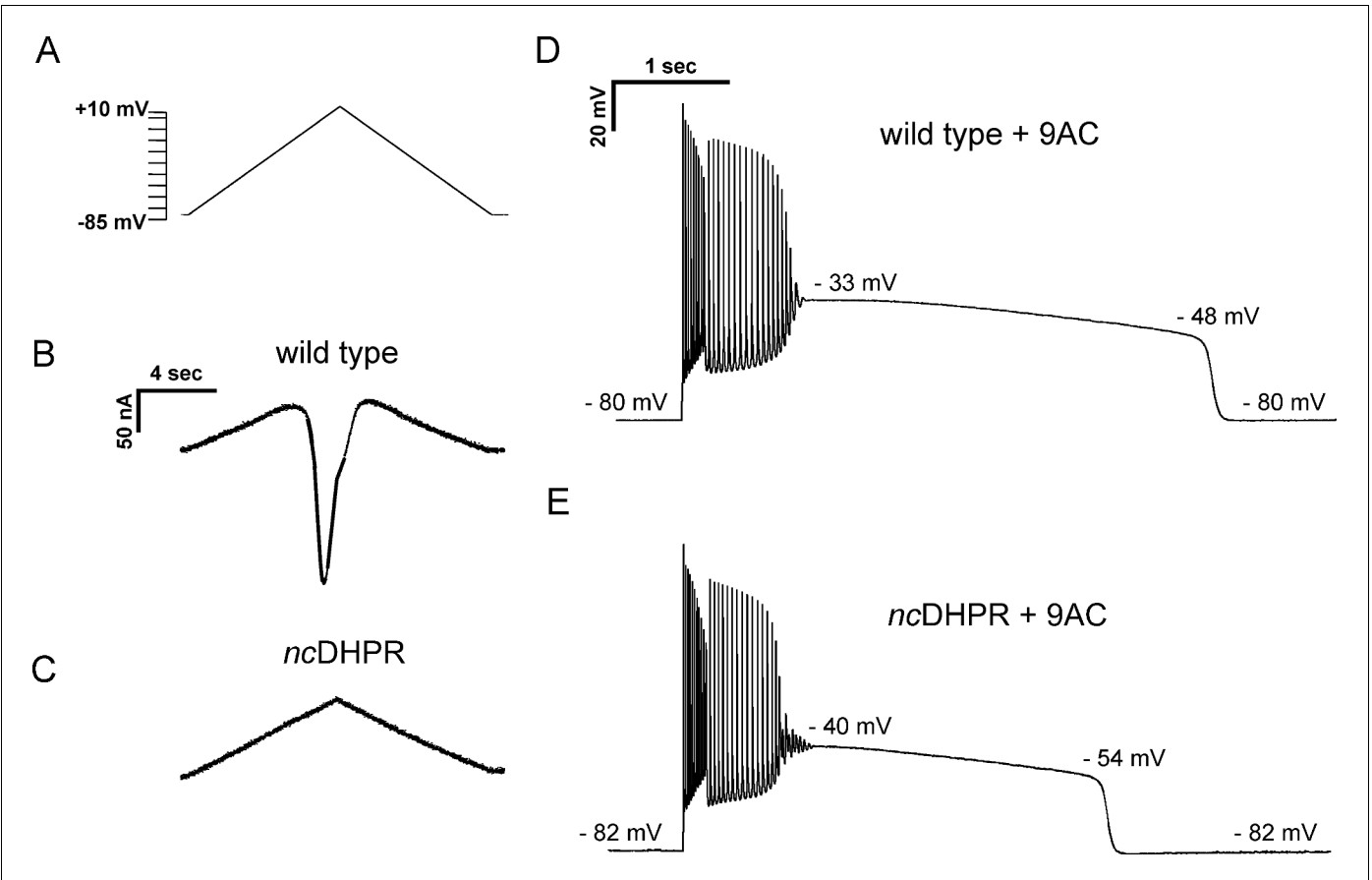

**Figure 6.** Current flow through Ca$_v$1.1 does not initiate, but helps to sustain, plateau potentials in 9-AC treated muscle. (A) The voltage protocol applied to FDB/IO fibers consisted of a ramp depolarization from −85 mV to +10 mV applied over 8 s. (B) A large inward current is present in wild-type muscle when Na$^+$, K$^+$, and Cl$^−$ currents were blocked. (C) ncDHPR muscle fibers confirm the absence of inward Ca$^{2+}$ current as there is only the linear change in current due to the changing command potential. (D) A plateau potential in a 9-AC treated wild-type muscle fiber. (E) A plateau potential in a 9-AC treated ncDHPR fiber.

tested if ranolazine prevented plateau potentials by hyperpolarizing the mean membrane potential prior to the development of plateau potentials. Untreated fibers with additional action potentials and plateau potentials occurring during the 2 s stimulation (myotonia) were excluded from the analysis to ensure that the presence of myotonia or plateau potentials did not account for the difference in mean membrane potential. As all untreated fibers initially had plateau potentials or

**Table 2.** Plateau potential parameters in 9-AC treated wild-type and ncDHPR muscle.

| | Pre-myotonia vm (mV) | Initial plateau vm (mV) | PP repol slope (mV/s) | Final PP vm (mV) | Max repol rate (mV/s) | Post PP vm (mV) | Duration of PP (s) |
|---|---|---|---|---|---|---|---|
| Wild type | −79.3 ± 0.9 (−79.5) | −36.0 ± 1.6 (−36.0) | −3.3 ± 0.7 (−3.1) | −48.1 ± 1.2 (−48.5) | −125 ± 6 (125) | −79.5 ± 0.4 (−79.7) | 3.9 ± 0.7 (3.8) |
| ncDHPR | −78.9 ± 0.6 (−79.0) | −37.2 ± 0.8 (−37.1) | −5.4 ± 0.6* (−5.3) | −49.2 ± 1.1 (−49.7) | −134 ± 10 (138) | −79 ± 0.8 (−79.3) | 2.4 ± 0.3* (2.3) |

Shown are the mean value and the standard deviation for parameters of plateau potentials in 9-AC treated wild-type and ncDHPR fibers. Below the mean value in parentheses is the median value. n = 41 fibers from four wild-type mice and n = 47 fibers from three ncDHPR mice. Vm is membrane potential, and repol is repolarization. Pre-myotonia Vm is the resting membrane potential prior to stimulation. Initial PP Vm is the membrane potential at the beginning of the plateau potential. Final PP Vm is the membrane potential prior to the sudden repolarization terminating the plateau potential. Max repol rate is the maximum rate of repolarization during the sudden repolarization. Post PP Vm is the membrane potential following termination of the plateau potential. * indicates p=0.01 for both significant differences (t-test).

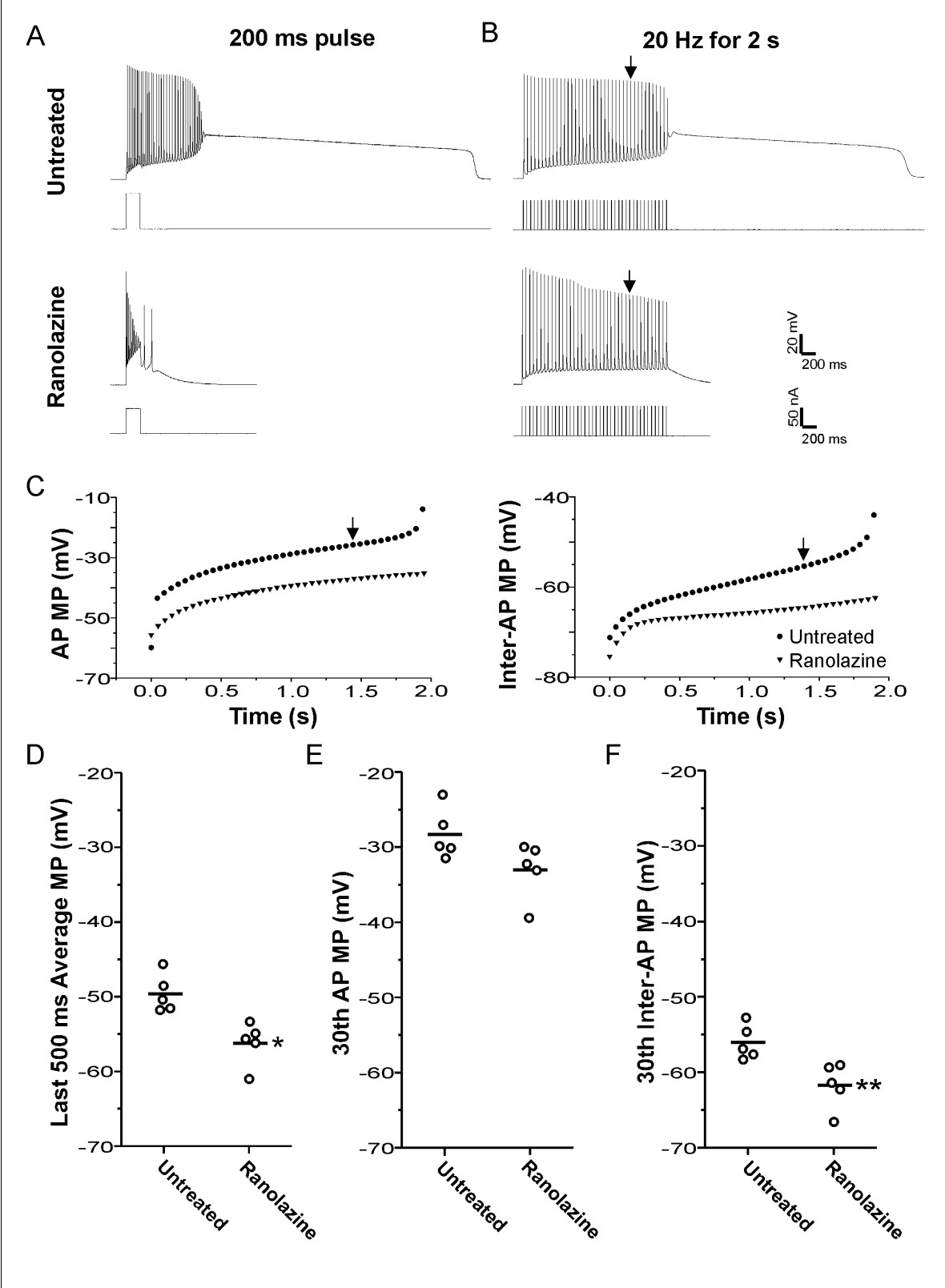

**Figure 7.** Block of plateau potentials by ranolazine in 9-AC treated muscle is associated with lessening of depolarization during repetitive firing. (**A**) Examples of the response to a 200 ms injection of depolarizing current at baseline and following treatment with 40 μM ranolazine. (**B**) Examples of the response to 2 s of stimulation at 20 Hz. The vertical arrow points to the 30th action potential in each trace, which was used for the analysis of differences between ranolazine untreated and ranolazine treated muscle (**D–F**). (**C**) Plots of the membrane potential during the 8 ms encompassing the

*Figure 7 continued on next page*

*Figure 7 continued*

action potential and the 42 ms encompassing the interspike interval for the untreated and ranolazine traces shown in **B**. Each point represents the value for a single action potential during the 2 s of 20 Hz stimulation. Action potential MP = the mean membrane potential during the 8 ms encompassing the action potential, Inter-AP MP = the mean membrane potential during the 42 ms interspike interval. (**D**) Plot of the mean membrane potential during the last 500 ms of stimulation. Each point represents an animal average derived from at least four fibers without (n = 5) and with treatment with ranolazine (n = 5). The horizontal line represents the mean. (**E and F**) Plots of the mean membrane potential during the 30th action potential and the 30th interspike interval. * indicates p=0.005 (t-test), **p=0.01 (t-test).

myotonia during repetitive stimulation, the analysis was not possible. To address this issue, the warm-up phenomenon was induced using 8 Hz 10 s trains, which lessens myotonia due to slow inactivation of Na channels (*Novak et al., 2015*). It must thus be noted that the untreated muscle was not at baseline but had already undergone some Na channel inactivation. No induction of warm-up was necessary following treatment with ranolazine, as no fibers had myotonia or plateau potentials. In the absence of ranolazine, the mean membrane potential during the 500 ms prior to the plateau potential was −49.6 ± 2.5 mV (n = 5 muscles, 28 fibers). In the presence of 40 µM ranolazine, the mean membrane potential was less depolarized (−56.2 ± 2.9 mV, p<0.05 vs untreated, n = 5 muscles, *Figure 7D*). These data suggest that ranolazine prevents plateau potentials by lessening depolarization of the mean membrane potential during the 20 Hz stimulation.

There are two contributors to the mean membrane potential during repetitive stimulation: (1) the membrane potential during action potentials, and (2) the membrane potential during the interspike interval. We analyzed the contribution of each of these to the hyperpolarization caused by ranolazine. By the 30th action potential of the 20 Hz stimulation, action potential duration had increased to close to 8 ms. This widening of the spike-form was likely due to failure of the membrane potential to fully repolarize between action potentials, given that depolarization has previously been found to cause widening of action potentials (*Renaud and Light, 1992*; *Yensen et al., 2002*; *Miranda et al., 2017*). We examined the mean membrane potential during the 8 ms encompassing the 30th action potential and found a trend (not statistically significant) toward lessening of depolarization following treatment with ranolazine (*Figure 7C,E*: −28.2 ± 3.4 vs −32.9 ± 3.8 mV, p=0.07). With 20 Hz stimulation, there is an action potential every 50 ms. After taking the mean membrane potential for the 8 ms encompassing the 30th action potential, there remained 42 ms in which there was no action potential. The membrane potential for this 42 ms interspike interval before the 30th action potential was less depolarized following treatment with ranolazine (*Figure 7C,F*: −56.0 ± 2.3 vs −61.7 ± 3.0 mV, p<0.05). As the interspike interval accounts for 84% (42/50ms) of the time between spikes during 20 Hz stimulation, hyperpolarization of this interval is largely responsible for hyperpolarization of the mean membrane potential.

## Ranolazine prevents transient weakness in vivo

The finding that ranolazine eliminates plateau potentials allowed us to explore whether plateau potentials are the mechanism underlying transient weakness in vivo. We recorded triceps surae force in 5 ClC[adr] myotonic mice before and 45 min after intraperitoneal (i.p.) injection of 50 mg/kg of ranolazine. As shown previously (*Novak et al., 2015*), treatment with ranolazine decreased myotonia such that muscle was able to more rapidly relax following termination of stimulation (*Figure 8A*). In addition to lessening myotonia, treatment with ranolazine eliminated transient weakness in all five mice (*Figure 8A and B*, p<0.01 vs untreated at 16 s). The elimination of both plateau potentials and transient weakness by ranolazine supports the hypothesis that plateau potentials are the mechanism underlying transient weakness in vivo.

## Discussion

Motor dysfunction in recessive myotonia (Becker disease) involves both muscle stiffness and transient weakness. Using intracellular recordings from a mouse model of myotonia congenita (ClC[adr]), we discovered that while some runs of myotonia resolved with repolarization, others terminated with a plateau potential; that is, depolarization to a membrane potential between −30 and −45 mV, lasting up to 100 s. There was gradual repolarization during plateau potentials until the membrane potential reached −45 mV, at which point a sudden repolarization to the resting membrane potential

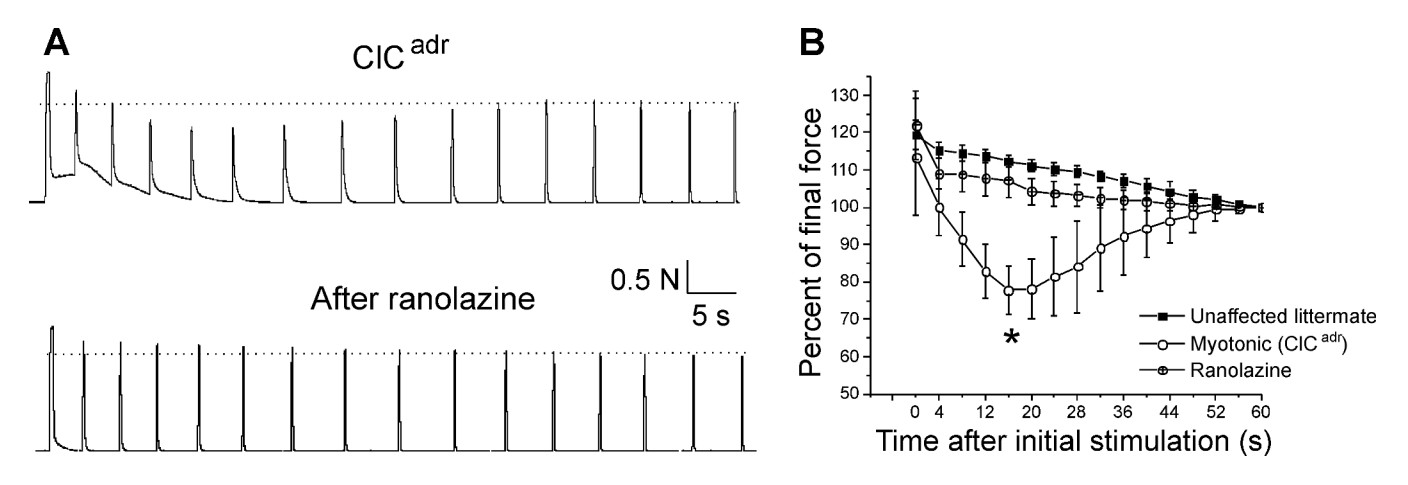

**Figure 8.** Ranolazine eliminates transient weakness in vivo. (A) Shown are the force traces from a triceps surae muscle group before and 45 min after i.p. injection of 50 mg/kg ranolazine. (B) Shown is the mean normalized force in 5 ClC[adr] mice before and after injection of ranolazine. p=0.001 for the difference in force 16 s (*) after the initial stimulation (paired t-test, 95% confidence interval 70–86 vs 102–113). The unaffected littermate data from *Figure 1* is replotted here to allow for comparison.

occurred. During plateau potentials, muscle fibers were inexcitable. Studies of genetic and pharmacologic mouse models suggest both current through voltage-activated $Ca_V1.1$ $Ca^{2+}$ channels and NaPIC may contribute to plateau potentials. NaPIC through $Na_V1.4$ channels is the key trigger of plateau potentials and current through $Ca_V1.1$ $Ca^{2+}$ channels contributes to sustaining plateau potentials. Blocking NaPIC with ranolazine eliminated both plateau potentials in vitro and transient weakness in vivo. Our results suggest that plateau potentials are the mechanism underlying transient weakness in Becker disease.

## Rapid transition between hyperexcitability and inexcitability in Becker disease

Our data suggest that muscle from a mouse model of Becker disease undergoes a rapid transition between the states of hyperexcitability (myotonia) and inexcitability (due to plateau potentials), shown in *Figure 9*. Repeated firing of action potentials during voluntary contraction triggers myotonia, which often transitions to a depolarization that forms a plateau potential. During plateau potentials, fibers cannot generate action potentials in response to stimulation, providing an explanation for the drop in CMAP amplitude reported in patients (*Ricker and Meinck, 1972*; *Brown, 1974*; *Aminoff et al., 1977*; *Deymeer et al., 1998*; *Drost et al., 2001*; *Modoni et al., 2011*). The reason for the rapid transition between myotonia (hyperexcitability) and plateau potentials (inexcitability) is that both states are caused by depolarization secondary to loss of muscle Cl⁻ current. The difference is one of degree: when depolarization is mild, Na⁺ channels are not inactivated, such that repetitive firing of action potentials is triggered. When depolarization worsens, Na⁺ channels inactivate, such that inexcitability and paralysis ensue. This proposal is similar to the current understanding of hyperkalemic periodic paralysis, in which there is often myotonia at the beginning of attacks (during the initial, mild depolarization) and weakness at the height of an attack (when depolarization is maximal) (*Cannon, 2015*; *Statland et al., 2018*).

## Ion channels contributing to generation and maintenance of plateau potentials

Involvement of voltage-gated channels in the generation of plateau potentials is suggested by the strong correlation between membrane potential and the initiation and termination of plateau potentials. In runs of myotonia that terminated in a plateau potential, the mean membrane potentials averaged to include both action potentials and interspike intervals prior to development of a plateau potential was −40 mV, whereas, within the same fibers, runs of myotonia terminating with repolarization had a mean membrane potential prior to repolarization of −49 mV. The midpoint of the

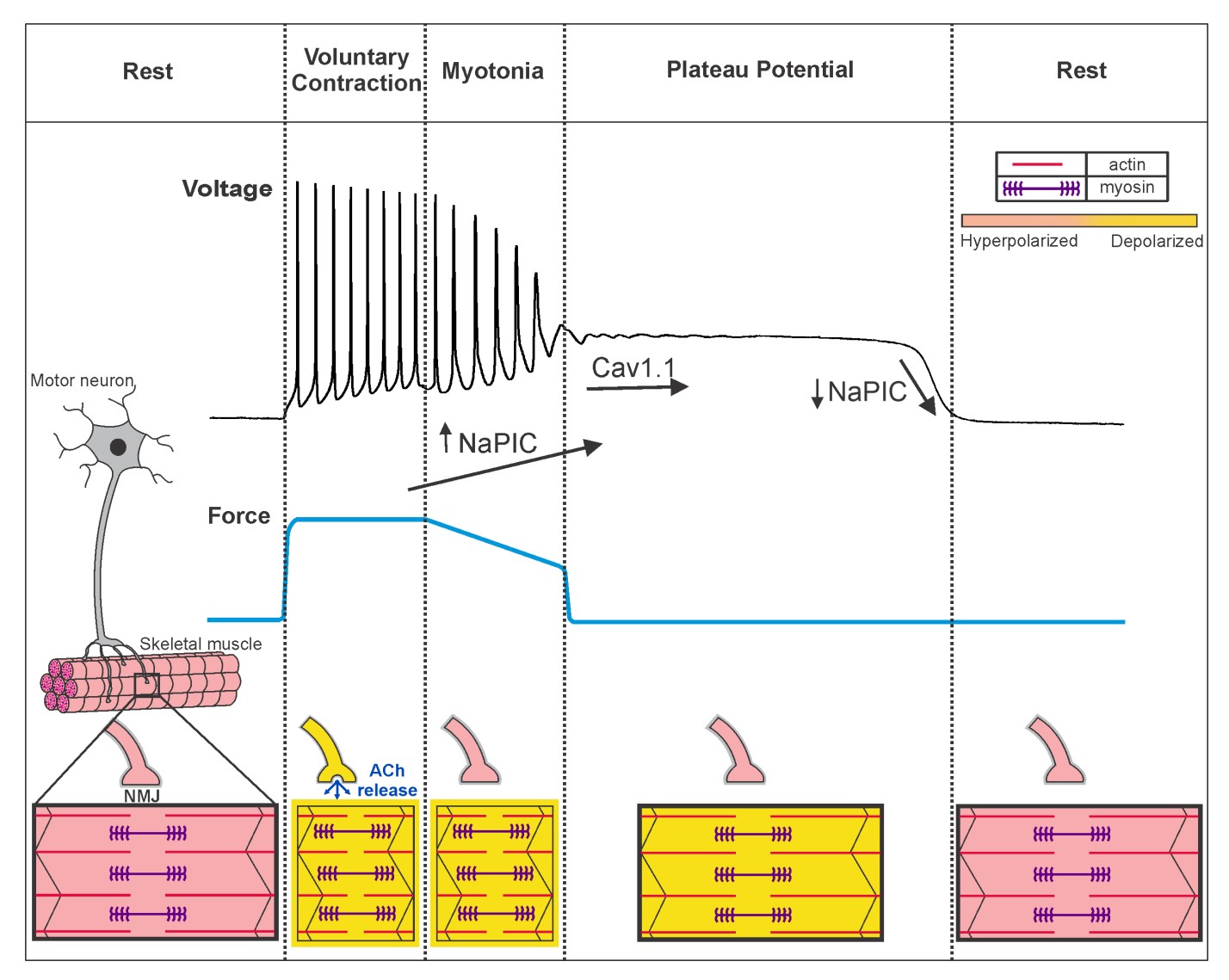

**Figure 9.** Voluntary contraction in myotonia congenita triggers a sequential progression through states of hyperexcitability and inexcitability. Shown on the left is a motor unit consisting of a motor neuron and the muscle fibers it innervates. At rest, both the motor neuron and muscle fibers are hyperpolarized and muscle is relaxed. To initiate voluntary contraction, the motor neuron fires repeated action potentials, which activate the neuromuscular junction (NMJ) to trigger repeated firing of action potentials in muscle (indicated by a yellow outline around the fiber). With the rapid firing of muscle action potentials, there is sustained contraction and force production (blue line). At the end of voluntary contraction, the motor neuron stops firing and the NMJ repolarizes, but in myotonic muscle there is continued involuntary firing of action potentials, which slows relaxation of muscle. When myotonia is terminated by transition into a plateau potential, muscle remains depolarized but cannot fire action potentials (black outline of the fiber), such that muscle is paralyzed. Finally, there is sudden repolarization and return of muscle to the resting state. Under the voltage trace our hypothesis regarding the contribution of NaPIC and current through $Ca_V1.1$ channels is illustrated. NaPIC begins to turn on at the end of voluntary contraction and triggers myotonia. In some runs of myotonia, NaPIC continues to increase and contributes to initiation of plateau potentials. The initial plateau potential voltage is sufficiently depolarized to trigger opening of $Ca_V1.1$ channels, which help to sustain plateau potentials. The gradual repolarization during plateau potentials causes $Ca_V1.1$ channels to close prior to termination of the potential. Rapid, voltage-dependent closing of $Na^+$ channels in the NaPIC mode contributes to termination of plateau potentials. ACh = acetylcholine.

difference between these values is close to −44 mV. In ClC[adr] muscle, the mean membrane potential prior to termination of plateau potentials was −46 mV, and in 9-AC treated muscle it was −45 mV. Thus, a mean membrane potential of −44 mV appears to predict both entry into, as well as termination of, plateau potentials.

Both Ca$_V$1.1 and Na$_V$1.4 are voltage-gated channels that could depolarize muscle to potentials achieved during plateau potentials. In spinal motor neurons, Ca$_V$1.3 channels play a central role in the generation of plateau potentials that last many seconds (*Alaburda et al., 2002*; *Heckman and Enoka, 2012*; *Hounsgaard, 2017*). Here, we show that the development of plateau potentials was not affected in *nc*DHPR muscle that lack Ca$^{2+}$ current through Ca$_V$1.1. However, there was a reduction in the duration of plateau potentials in *nc*DHPR muscle. These data suggest that Ca$^{2+}$ flux through Ca$_V$1.1 channels contributes to sustaining plateau potentials. This was a surprise, as the membrane potential during plateau potentials is more negative than voltages at which there is significant current through Ca$_V$1.1 channels (*García and Beam, 1994*; *Bannister and Beam, 2013*). One explanation is that in intact, mature fibers, prolonged depolarization during plateau potentials allows for activation of Ca$_V$1.1 at more negative potentials than during the shorter step depolarizations typically used for voltage clamp studies. Our findings suggest that despite Ca$^{2+}$ current through Ca$_V$1.1 having no essential role in healthy muscle (*Dayal et al., 2017*), it may contribute to pathologic depolarization in muscle channelopathies. It is not clear at this point whether the primary role of Ca$^{2+}$ flux through Ca$_V$1.1 channels in prolonging plateau potentials is a direct effect of increased Ca$^{2+}$ conductance or whether it is due to effects of Ca$^{2+}$ accumulation on other ion channels.

Partial block of Na$_V$1.4 channels with ranolazine eliminated plateau potentials, suggesting involvement of Na$_V$1.4. However, the majority of Na$_V$1.4 channels inactivate within ms (fast-inactivating Na$_V$1.4 channels), such that they cannot contribute to prolonged depolarization during plateau potentials. There is a NaPIC in skeletal muscle that appears to arise from modal gating whereby, with very low frequency, any Na$_V$1.4 channel may enter the non-inactivating mode (*Patlak and Ortiz, 1986*; *Gage et al., 1989*; *Hawash et al., 2017*). The subset of Na$_V$1.4 channels in the non-activating mode appears to play a role in the development of plateau potentials.

Ranolazine has been found to preferentially block NaPIC (*El-Bizri et al., 2011*; *Kahlig et al., 2014*). When myotonic muscle was treated with ranolazine to block NaPIC, depolarization of the mean membrane potential during repetitive firing was decreased and plateau potentials were prevented. An important contributor was lessening of depolarization of the membrane potential during the interspike interval. Previously, the only identified contributor to depolarization of the interspike membrane potential was K$^+$ build-up in the transverse (t)-tubules (invaginations of the sarcolemma), which depolarizes the K$^+$ equilibrium potential (*Adrian and Bryant, 1974*; *Adrian and Marshall, 1976*; *Wallinga et al., 1999*; *Fraser et al., 2011*). The finding that block of NaPIC lessens depolarization of the interspike membrane potential suggests that NaPIC is activated during this interval.

We hypothesize that activation of NaPIC combines with K$^+$ build-up in t-tubules to depolarize muscle to a mean membrane potential of −44 mV during myotonia, such that a plateau potential is triggered. An additional contributor to this depolarization may be the lessening of inward rectifier potassium channel (Kir) conductance with depolarization (*Standen and Stanfield, 1980*; *Struyk and Cannon, 2008*). It is unclear whether NaPIC, K$^+$ build-up, and decreased Kir conductance can fully account for depolarization during plateau potentials. Based on the studies of sustained depolarizations (sometimes termed plateau potentials) in neurons, a family of ion channels that might contribute is the transient receptor potential (TRP) ion channel family (*Yan et al., 2009*; *Phelan et al., 2012*). Members of the TRP ion channel family are expressed in skeletal muscle (*Brinkmeier, 2011*; *Gailly, 2012*), and we recently found that activation of TRPV4 plays a role in triggering percussion myotonia (*Dupont et al., 2020*).

While entry into plateau potentials was often gradual (*Figure 4*), it could also occur over two to four action potentials (~100–200 ms, *Figure 2*, *Figure 4*). Repolarization occurred over 200–300 ms. While we do not know all the channels involved, we hypothesize one contributor to the instances of rapid onset and termination of plateau potentials is activation and deactivation of NaPIC (*Figure 9*). NaPIC in muscle can activate and deactivate within 10 ms (*Gage et al., 1989*). The impression that gating of NaPIC is slow comes from the slow ramp protocols often used to study NaPIC. These protocols enable inactivation of fast Na current such that NaPIC can be studied in isolation (*Hawash et al., 2017*).

Ion channels promoting repolarization are likely also involved in the sudden termination of plateau potentials. Oscillations in membrane potential often occurred at the beginning or end of plateau potentials. In motor neurons, oscillations in membrane potential are caused by a balance between two voltage-gated ion channels: one promoting depolarization and one promoting

hyperpolarization (*Iglesias et al., 2011*; *Sciamanna and Wilson, 2011*; *Nardelli et al., 2017*). $K_v$ channels could participate in oscillations occurring during plateau potentials given that $K_v$ conductance in muscle is large and the channels begin to activate at voltages reached during plateau potentials (*Beam and Donaldson, 1983*; *DiFranco et al., 2012*). Thus, it may be possible to prevent plateau potentials by combining partial block of channels promoting depolarization such as NaPIC with partial opening of channels promoting repolarization such as $K_v$ channels.

As an example of how such an approach might work, plateau potentials in 9AC treated muscle were significantly shorter than in ClC$^{adr}$ muscle. Two factors may explain the difference. The first is that ClC-1 conductance is not completely blocked following application of 100 µM 9-AC (*Palade and Barchi, 1977*). While small, the remaining current through ClC-1 channels promotes repolarization and shortens plateau potentials. The second is that NaPIC current is larger in ClC$^{adr}$ muscle compared to wild-type muscle (used for the 9-AC model of myotonia) (*Hawash et al., 2017*). The combination of slightly higher ClC-1 conductance and slightly smaller NaPIC likely combine to shorten plateau potentials in the 9-AC model of myotonia. However, despite the shorter duration of plateau potentials in the 9-AC model of myotonia, they were more frequent (92% of fibers vs 30% of fibers). The reason for this difference is not known and raises the possibility that currents not yet identified play a role in development of plateau potentials.

### Blocking NaPIC as therapy for transient weakness

In a mouse model of myotonia congenita, ranolazine prevents both development of plateau potentials in vitro and transient weakness in vivo. In open label trials of ranolazine in both myotonia congenita and paramyotonia congenita, there were statistically significant reductions in the degree of self-reported weakness (*Arnold et al., 2017*; *Lorusso et al., 2019*). Taken together, these data suggest that the clinical benefit of blocking $Na^+$ channels in some diseases with myotonia may result, in part, from prevention of transient weakness secondary to development of plateau potentials. Our data also raises the possibility that blocking of Cav1.1 channels might reduce transient weakness by shortening the duration of plateau potentials. Since blockers of L-type $Ca^{2+}$ channels are used clinically to treat hypertension and have few side effects, a trial of their efficacy in reducing transient weakness may be worthwhile.

Mutations of $Na_V1.4$ responsible for hyperkalemic periodic paralysis increase NaPIC (*Cannon et al., 1991*; *Cannon and Strittmatter, 1993a*), which appears to play a central role in triggering the depolarization that underlies attacks of transient weakness (*Lehmann-Horn et al., 1987*; *Jurkat-Rott et al., 2010*; *Cannon, 2015*). This suggests that blocking NaPIC should be effective in treating hyperkalemic periodic paralysis. However, treating patients with $Na^+$ channel blockers that are effective in treating myotonia, such as mexiletine, was previously found to be ineffective (*Ricker et al., 1983*; *Ricker et al., 1986*). The finding that blocking NaPIC with ranolazine lessens depolarization of the interspike membrane potential suggests it might be worth considering a trial of this FDA-approved drug's efficacy in hyperkalemic periodic paralysis.

### Conclusion

We identified currents contributing to plateau potentials that are responsible for transient weakness in recessive myotonia congenita. We also determined that blocking a current contributing to plateau potentials provided effective amelioration of transient weakness. Currents contributing to plateau potentials are present in wild-type muscle; thus, they might contribute to depolarization in other muscle channelopathies with transient weakness, such as hyper- and hypokalemic periodic paralysis. Identification of channels involved in generation of plateau potentials in skeletal muscle may thus advance understanding of regulation of excitability in both healthy and diseased muscle.

## Materials and methods

**Key resources table**

| Reagent type (species) or resource | Designation | Source or reference | Identifiers | Additional information |
| --- | --- | --- | --- | --- |
| Strain, strain background (*Mus musculus*) | *Clcn1*$^{adr-mto}$/J (ClC$^{adr}$) mice | Jackson Labs | Stock #000939 | |

*Continued on next page*

Continued

| Reagent type (species) or resource | Designation | Source or reference | Identifiers | Additional information |
|---|---|---|---|---|
| Strain, strain background (*Mus musculus*) | *nc*DHPR mice | *Dayal et al., 2017* Nat Commun 8:475. | | |
| Chemical compound, drug | 9-Anthracene-carboxylic acid (9-AC) | Sigma | Cat. #: A4678 | 0.1 mM |
| Chemical compound, drug | N-benzyl-*p*-toluenesulfonamide (BTS) | TCI America | Prod. #: B3082 | 0.05 mM |
| Chemical compound, drug | Ranolazine | Sigma-Aldrich | Cat. #: R6152 | 0.04 mM |
| Chemical compound, drug | Tetrodotoxin (TTX) | Tocris | Cat. #: 1069 | 0.001 mM |
| Chemical compound, drug | 3,4-diaminopyridine (3,4-DAP) | Sigma-Aldrich | Cat. #: D7148 | 0.1 mM |
| Chemical compound, drug | Ouabain | Sigma-Aldrich | Cat. #: 03125 | 0.01 mM |
| Software, algorithm | Spike2 | http://ced.co.uk/downloads/latestsoftware | | Version 8 |
| Software, algorithm | MATLAB | https://www.mathworks.com/downloads | | |

## Mice

All animal procedures were performed in accordance with the policies of the Animal Care and Use Committee of Wright State University and were conducted in accordance with the United States Public Health Service's Policy on Humane Care and Use of Laboratory Animals.

The genetic mouse model of myotonia congenita used was $Clcn1^{adr-mto}$/J (ClC$^{adr}$) mice, which have a homozygous null mutation in the *Clcn1* gene (Jackson Laboratory Stock #000939). The pharmacologic model of Becker disease involved treatment of muscle with 100 μM 9-anthracenecarboxylic acid (9-AC). The mouse model of $Ca^{2+}$ non-conducting $Ca_V1.1$ used was *nc*DHPR, carrying a point mutation in the *Cacna1S* gene coding for N617D in pore loop II (*Dayal et al., 2017*).

Genotyping of ClC$^{adr}$ mice was performed as previously described to select heterozygous mice for breeding (*Dupont et al., 2019*). Otherwise, homozygous myotonic mice were identified by appearance and behavior as previously described (*Novak et al., 2015*). Unaffected littermates were used as controls. Genotyping for selection of homozygous *nc*DHPR was performed as previously described (*Dayal et al., 2017*). Both male and female mice were used from 2 months to 6 months of age. As mice with myotonia have difficulty climbing to reach food, symptomatic mice were supplied with moistened chow paste (Irradiated Rodent Diet; Harlan Teklad 2918) on the floor of the cage.

## Electrophysiology

Current and voltage clamp recordings were performed at 20–22°C.

## Current clamp

Mice were sacrificed using $CO_2$ inhalation followed by cervical dislocation, and both extensor digitorum longus (EDL) muscles were dissected out tendon-to-tendon. Muscles were maintained and recorded at 22°C within 6 hr of sacrifice. The recording chamber was continuously perfused with Ringer solution containing (in mM) NaCl, 118; KCl, 3.5; $CaCl_2$, 1.5; $MgSO_4$, 0.7; $NaHCO_3$, 26.2; $NaH_2PO_4$, 1.7; glucose, 5.5 (pH 7.3–7.4 at 20–22°C), and equilibrated with 95% $O_2$ and 5% $CO_2$.

Intracellular recordings were performed as previously described (*Novak et al., 2015*; *Hawash et al., 2017*; *Dupont et al., 2019*). Briefly, muscles were loaded with 50 μM N-benzyl-p-toluenesulfonamide (BTS, Tokyo Chemical Industry) for at least 30 min prior to recording to prevent contraction. BTS was dissolved in DMSO and added to the perfusate prior to bubbling with 95% $O_2$ and 5% $CO_2$, as we have found that BTS is more soluble at a basic pH. Prior to recording, muscles were stained for 3 min with 10 μM 4-(4-diethylaminostyrl)-N-methylpyridinium iodide (4-Di-2ASP, Molecular Probes) to allow imaging muscle with an upright epifluorescence microscope (Leica DMR, Bannockburn, IL).

Micro-electrodes were filled with 3M KCl solution containing 1 mM sulforhodamine to visualize the electrodes with epifluorescence. Resistances were between 15 and 30 MΩ, and capacitance

compensation was optimized prior to recording. Fibers with resting potentials more depolarized than –74 mV were excluded from analysis.

In cases where there was failure of action potentials during trains of stimulation, we did not determine whether the failure was due to the presence of an absolute or relative refractory period by altering current injection during the train of stimuli. Successful repolarization following a plateau potential was defined as a return to within 4 mV of the resting potential prior to the plateau potential. Fibers that did not fully repolarize may have become damaged and were thus discarded from further analysis.

## Voltage clamp

*Flexor digitorum brevis* (FDB) and *interosseous* (IO) muscle fibers were isolated as previously described (*Waters et al., 2013*; *Hawash et al., 2017*). Briefly, muscles were surgically removed and enzymatically dissociated at 37°C under mild agitation for ~1 hr using 1000 U/mL of collagenase type IV (Worthington Biochemical). Mechanical dissociation was completed using mild trituration in buffer with no collagenase. The fibers were allowed to recover at 20–22°C for 1 hr before being used for electrical measurements.

Both the current-passing and voltage-sensing electrodes were filled with internal solution (see below) and had resistances of ~15 M$\Omega$. After impalement, 10 min of hyperpolarizing current injection was allowed for equilibration of the electrode solution. Data were acquired at 20 kHz and low-pass filtered with the internal Axoclamp 900A filters at 1 kHz. The voltage clamp command signal was low-pass filtered with an external Warner LFP-8 at 1 kHz.

Internal solution (in mM) was as follows: 75 aspartate, 30 EGTA, 15 Ca(OH)$_2$, 5 MgCl$_2$, 5 ATP di-Na, five phosphocreatine di-Na, five glutathione, 20 MOPS, and pH 7.2 with CsOH.

Extracellular solution (in mM) was as follows: 144 NaCl, 4 CsCl, 1.2 CaCl$_2$, 0.6 MgCl$_2$, five glucose, 1 NaH$_2$PO$_4$, 10 MOPS, 0.05 BaCl$_2$, 0.1 9-AC, 0.001 TTX, 0.01 Ouabain, 0.1 3,4-diaminopyridine (3,4-DAP), and pH 7.4 with NaOH.

## In vivo muscle force recording

In vivo muscle force recordings were performed as previously described (*Dupont et al., 2019*; *Wang et al., 2020*). Briefly, mice were anesthetized via isoflurane inhalation; then the distal tendon of the triceps surae muscles was attached to a force transduction motor and the sciatic nerve was stimulated while isometric muscle force generation was measured. The sciatic nerve was stimulated with constant current injection. The amplitude of the current pulse was adjusted to 150% of the current required to trigger a single action potential. To induce myotonia, 45 pulses of 1 ms duration were delivered at 100 Hz. To follow the development of transient weakness, 15 pulses delivered at 100 Hz were delivered every 4 s. Muscle temperature was monitored with a laser probe and maintained between 29°C and 31°C with a heat lamp. The muscle was kept moist by applying mineral oil. Ranolazine was administered via intraperitoneal (i.p.) injection at a dose of 50 mg/kg dissolved in water. The typical volume of water injected was 100 µl.

## Statistics

Sample size was determined by past practice, where we have found that an n of 5 muscles from five different mice, studied on different days, yields statistically significant differences in muscle action potential properties. At least five muscle fibers were recorded from each muscle. However, for studies of plateau potentials in ClC$^{adr}$ mice we obtained recordings with adequate preservation of resting membrane potential in only a subset of fibers, so fewer fibers per muscle were included in the final analysis. This was due to the prolonged duration of plateau potentials, which required maintaining prolonged impalement of individual fibers with two electrodes. No outlier data points were excluded. Intracellular recording data from different mice were analyzed using nested analysis of variance with n as the number of mice, with data presented as mean ± SD. $p < 0.05$ was considered to be significant. The numbers of animals and fibers used are described in the corresponding figure legends and text. For parameters that were not normally distributed, such as duration of the plateau potential and slope of the repolarization, differences between two data sets were analyzed after applying a log transformation, which yielded normally distributed data.

For comparisons of data within individual muscle fibers (mean membrane potential and mean firing rate for runs of myotonia, without and with plateau potentials), the paired Student's t-test was used with n as the number of fibers. For force recording comparisons before and after ranolazine treatment, the paired Student's t-test was used with n as the number of mice. For comparisons of force recordings between myotonic mice and unaffected littermates, a two-sample Student's t-test was used.

## Acknowledgements

This work was supported by NIH grant AR074985 (MMR), MDA grant 602459 (MMR), and Austrian Science Fund (FWF) Grants P23229 (MG) and P27392 (MG and AD).

## Additional information

### Competing interests

Kevin R Novak: is an employee of Evokes LLC. The author has no other competing interests to declare. The other authors declare that no competing interests exist.

### Funding

| Funder | Grant reference number | Author |
|---|---|---|
| National Institutes of Health | AR074985 | Mark M Rich |
| Muscular Dystrophy Association | 602459 | Mark M Rich |
| Austrian Science Fund | P23229 | Manfred Grabner |
| Austrian Science Fund | P27392 | Manfred Grabner Anamika Dayal |

The funders had no role in study design, data collection and interpretation, or the decision to submit the work for publication.

### Author contributions

Jessica H Myers, Software, Formal analysis, Investigation; Kirsten Denman, Chris DuPont, Ahmed A Hawash, Kevin R Novak, Andrew Koesters, Formal analysis, Investigation; Manfred Grabner, Anamika Dayal, Resources, Writing - review and editing; Andrew A Voss, Resources, Methodology, Writing - review and editing; Mark M Rich, Conceptualization, Resources, Data curation, Formal analysis, Supervision, Funding acquisition, Writing - original draft, Writing - review and editing

### Author ORCIDs

Andrew Koesters http://orcid.org/0000-0003-3281-188X
Manfred Grabner http://orcid.org/0000-0002-5196-4024
Anamika Dayal http://orcid.org/0000-0001-8075-8812
Mark M Rich https://orcid.org/0000-0002-6956-5531

### Ethics

Animal experimentation: This study was performed in strict accordance with the recommendations in the Guide for the Care and Use of Laboratory Animals of the National Institutes of Health. All of the animals were handled according to approved institutional animal care and use committee (IACUC) protocols 1179 and 1188, Wright State University.

### Decision letter and Author response

Decision letter https://doi.org/10.7554/eLife.65691.sa1
Author response https://doi.org/10.7554/eLife.65691.sa2

# Additional files

## Supplementary files

• Transparent reporting form

## Data availability

We have uploaded our data to Dyrad: https://doi.org/10.5061/dryad.bvq83bk7q.

The following dataset was generated:

| Author(s) | Year | Dataset title | Dataset URL | Database and Identifier |
|---|---|---|---|---|
| Myers J, Rich M | 2021 | The mechanism underlying transient weakness in myotonia congenita | https://doi.org/10.5061/dryad.bvq83bk7q | Dryad Digital Repository, 10.5061/dryad.bvq83bk7q |

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
