## [Decision Letter]

**Acceptance summary:**

Patients with myotonia congenita (or Becker disease) experience episodes of transient weakness, but the mechanisms underlying this phenomenon are unknown. This study provides the most definitive experimental evidence to date for the mechanistic basis of transient weakness in myotonia congenita and also suggests ranolazine may be beneficial for prophylactic management. The results will be of interest to both muscle physiologists and clinicians that treat patients with myotonia congenita and other disorders of muscle hyperexcitability.

**Decision letter after peer review:**

Thank you for submitting your article "The mechanism underlying transient weakness in myotonia congenita" for consideration by *eLife*. Your article has been reviewed by 3 peer reviewers, including Henry M Colecraft as the Reviewing Editor and Reviewer #1, and the evaluation has been overseen by Richard Aldrich as the Senior Editor. The following individuals involved in review of your submission have agreed to reveal their identity: Robert T Dirksen (Reviewer #2); Stephen Cannon (Reviewer #3).

Essential Revisions:

1. Provide more detailed information regarding the criteria used to define successful return to the resting potential, the frequency in which fibers failed to repolarize to the resting potential (and thus were excluded from further analyses), and whether this was different between WT, adr, and 9-AC experiments

2. Briefer duration of plateau potentials (and steeper negative slope) observed with 9-AC treatment compared to adr mouse. The possibility that this could be due to incomplete CLC-1 inhibition by 9-AC compared to adr mouse should be included in Discussion..

3. A more complete discussion of the potential role of intracellular Ca^2+^-dependent mechanisms in controlling PP duration is warranted. The role of Ca^2+^-dependent processes can be parsed out under experimental conditions in which intracellular Ca^2+^ is buffered with BAPTA-am as suggested by Reviewer 2.

4. The summary figure (Figure 7) should be modified to include information regarding the proposed role of NAPIC and CaV1.1 conductances in PP incidence and duration, respectively.

Reviewer #1 (Recommendations for the authors):

1. It is unclear whether the level of NAPIC is increased in ClC-1 null mice or with pharmacological blockade of ClC-1 in WT mice. Voltage clamp experiments to directly compare the levels of NAPIC in control and ClC-1-null muscle fibers would clarify this point and strengthen the manuscript.

Reviewer #2 (Recommendations for the authors):

The data are of high quality, the results are intriguing, and the implications are of high importance to the field. The manuscript would be strengthened by the authors addressing the following issues.

1. In Figure 2E, the authors use short depolarizing current injections to monitor changes in membrane resistance during both sustained plateau potentials (PPs) and at the at the resting membrane potential (RMP) during the refractor period. From the representative trace shown in Figure 2E, it appears that the membrane potential becomes progressively more negative and the membrane resistance gradually increases during the PP. Together, these results are consistent with the closing of channels that produce inward/depolarizing current during the PP (e.g. Na/Ca channels), rather than opening of channels that produce an outward current (e.g. K channels). Is a similar pattern observed in other experiments? It would be helpful if the authors could carefully quantify average change in membrane resistance during the PP (or at least compare membrane resistance between early and late PP and during the RMP phase across all fibers). Indeed, such an analysis could provide additional support for the authors' proposed mechanism of termination of the PP.

2. The authors should speculate on or discuss the change in ion conductance(s) that likely underlie the relatively rapid repolarization to the resting potential observed at the end of the PP.

3. One observation that is not adequately discussed is the reason for why the PP duration in the acute 9AC model is so much briefer than that observed in the constitutive adr mouse model (e.g. Figure 3B)? This observation suggests, as hinted by the authors, that there are indeed compensatory mechanisms in Adr mice that increase PP duration. As one possibility, sustained NaPIC and/or CaV1.1 currents may be upregulated in Adr mice. In this regard, how does the magnitude of the PP resistance (which likely depends on PP inward currents as discussed above) compare between Adr mice and WT mice treated with 9AC? Is there any evidence for a similar increase in NaPIC and/or Cav1.1 currents in humans with myotonia congenital?

4. Since the incidence of PPs correlates with firing frequency during myotonia, the authors propose that increases in intracellular calcium could modulate the activity of channels responsible for the PP (lines 179-180). However, it not clear why this could not simply reflect voltage dependent changes in channel activity produced during the myotonia. Alternatively, higher firing rates would produce greater K accumulation and TT depolarization, which could also contribute to the observed correlation between firing frequency and incidence of PPs. In order to support the authors' preferred suggestion for a role of calcium-dependent mechanisms, experiments to more directly test this idea are needed (e.g. effect BAPTA-AM loading on firing rate, membrane resistance and plateau duration).

5. In Figure 6B, it would be helpful if the authors could also replot the untreated WT force data from Figure 1E as this would allow readers to more directly assess the relative degree of rescue by ranolazine treatment compared to control WT mice.

6. Unfortunately, force measurements in non-conducting Cav1.1. mice were not performed. Nevertheless, given the observed effect of Cav1.1 conductance on both PP duration and slope of repolarization as shown in Table 2, the authors should at least discuss the potential importance of Cav1.1 conductance in the transient weakness in myotonia congenita in the second to last paragraph of the Discussion.

7. The summary figure presented in the Discussion section (Figure 7) should also include information regarding the proposed role of NaPIC and CaV1.1 conductances in PP incidence and duration, respectively, which is the main contribution of this study.

Reviewer #3 (Recommendations for the authors):

1. Page 14, line 291, perhaps state "by hyperpolarizing the mean membrane potential". The interpretation of a "decreasing" membrane potential can be ambiguous. Decreasing toward 0 mV (less polarized) or decreasing meaning more negative value (hyperpolarized).

2. Page 20, line 441, "… a subset of NaV1.4 channels that lack fast inactivation". This gives the impression of two separate functional classes of NaV1.4 channels: those the fast inactivate (the overwhelming majority) and that do not fast inactivate (a small minority). Instead, Dr. Patlak (1986) proposed modal gating, whereby with very low frequency any NaV1.4 channel may enter the non-inactivating mode, and then subsequently convert back. This remains the prevailing view.

---

## [Author Response]

Essential Revisions:1. Provide more detailed information regarding the criteria used to define successful return to the resting potential, the frequency in which fibers failed to repolarize to the resting potential (and thus were excluded from further analyses), and whether this was different between WT, adr, and 9-AC experiments

This is an excellent point, which made us critically re-evaluate our data. In some of our early recordings of plateau potentials in ClCadr muscle, the repolarization was not as complete as in our recent recordings. We believe this difference is due to our improved recording techniques. We thus eliminated our early recordings of plateau potentials from the data set used for analyses. We performed new intracellular recordings from ClCadr muscle fibers, which we added to the data set. The criteria for successful return to resting potential was repolarization to within 4 mV of the original resting potential. We have replaced the data for ClCadr mice in Table 1 with the new data and have also re-made Figure 3B with the new data. Text describing the criterion for repolarization has been added to the methods.

We have done the requested analysis of the frequency of lack of repolarization in ClCadr and 9-AC treated muscle. In ClCadr mice we recorded from 119 fibers. 36 had plateau potentials. Of those, 26 repolarized to within 4 mV of the original resting potential. Following treatment of WT fibers with 9AC, all plateau potentials ended with repolarization to within 4 mV of the resting potential (49/49 fibers). Text describing the frequency of lack of repolarization is now included in the Results section. Plateau potentials were never observed in WT fibers.

2. Briefer duration of plateau potentials (and steeper negative slope) observed with 9-AC treatment compared to adr mouse. The possibility that this could be due to incomplete CLC-1 inhibition by 9-AC compared to adr mouse should be included in Discussion..

This is an excellent point and on further reflection, we agree that it is likely a major contributor to the difference. We have added this point to the discussion.

3. A more complete discussion of the potential role of intracellular Ca^2+^-dependent mechanisms in controlling PP duration is warranted. The role of Ca^2+^-dependent processes can be parsed out under experimental conditions in which intracellular Ca^2+^ is buffered with BAPTA-am as suggested by Reviewer 2.

Study of the contribution of Ca^2+^-dependent mechanisms to generation of plateau potentials is something that is of great interest. We have added a more complete discussion of the role of intracellular Ca^2+^ as requested. In the future we hope to begin experiments to further elucidate the role of intracellular Ca^2+^-dependent mechanisms.

4. The summary figure (Figure 7) should be modified to include information regarding the proposed role of NAPIC and CaV1.1 conductances in PP incidence and duration, respectively.

An excellent suggestion that improves the value of the figure. The requested modification has been made in the figure (now Figure 9).

Reviewer #1 (Recommendations for the authors):1. It is unclear whether the level of NAPIC is increased in ClC-1 null mice or with pharmacological blockade of ClC-1 in WT mice. Voltage clamp experiments to directly compare the levels of NAPIC in control and ClC-1-null muscle fibers would clarify this point and strengthen the manuscript.

This is an excellent point and an increase in NaPIC could explain the increased duration of plateau potentials in ClC-1 null mice. We previously compared NaPIC density in ClC-1 null fibers to control fibers and found a higher density in ClC-1 null mice (1.4 ± 0.2 nA/nF vs 0.9 ± 0.2 nA/nF) (Hawash et al., 2017). We have now added this point to a new Discussion section regarding the difference in duration of plateau potentials between ClC-1 null mice and 9-AC treated muscle. The difference in NaPIC in combination with incomplete block of ClC-1 channels with 9-AC likely explains the difference in duration of plateau potentials between the two groups.

Reviewer #2 (Recommendations for the authors):The data are of high quality, the results are intriguing, and the implications are of high importance to the field. The manuscript would be strengthened by the authors addressing the following issues.1. In Figure 2E, the authors use short depolarizing current injections to monitor changes in membrane resistance during both sustained plateau potentials (PPs) and at the at the resting membrane potential (RMP) during the refractor period. From the representative trace shown in Figure 2E, it appears that the membrane potential becomes progressively more negative and the membrane resistance gradually increases during the PP. Together, these results are consistent with the closing of channels that produce inward/depolarizing current during the PP (e.g. Na/Ca channels), rather than opening of channels that produce an outward current (e.g. K channels). Is a similar pattern observed in other experiments? It would be helpful if the authors could carefully quantify average change in membrane resistance during the PP (or at least compare membrane resistance between early and late PP and during the RMP phase across all fibers). Indeed, such an analysis could provide additional support for the authors' proposed mechanism of termination of the PP.

We have performed the requested experiments and added the results to the manuscript as a new figure (Figure 5). The response to current injection late in plateau potentials (just prior to repolarization) is clearly not passive as injection of current induces oscillations in membrane potential. It seems likely that one contributor to the oscillations following current injection during the late phase of plateau potentials is voltage dependent modulation of Kv channel conductance. This is now mentioned in the manuscript. Because of this we could not measure input resistance throughout plateau potentials. However, the response to current injection during the early phase of plateau potentials appeared passive, allowing us to measure input resistance. There was a statistically significant reduction in estimated input resistance during the early phase of plateau potentials, strongly suggesting there is a net increase in membrane conductance at that time.

2. The authors should speculate on or discuss the change in ion conductance(s) that likely underlie the relatively rapid repolarization to the resting potential observed at the end of the PP.

Consideration of potential contributors to the rapid depolarization is now included in the discussion and in the new version of the summary figure (now Figure 9).

3. One observation that is not adequately discussed is the reason for why the PP duration in the acute 9AC model is so much briefer than that observed in the constitutive adr mouse model (e.g. Figure 3B)? This observation suggests, as hinted by the authors, that there are indeed compensatory mechanisms in Adr mice that increase PP duration. As one possibility, sustained NaPIC and/or CaV1.1 currents may be upregulatedin Adr mice. In this regard, how does the magnitude of the PP resistance (which likely depends on PP inward currents as discussed above) compare between Adr mice and WT mice treated with 9AC? Is there any evidence for a similar increase in NaPIC and/or Cav1.1 currents in humans with myotonia congenital?

We have added a section to the discussion outlining two likely contributors to longer PP duration in ClCadr mice: increased NaPIC and lower (absent) ClC-1 conductance. Our previous finding that NaPIC is higher in ClCadr muscle has been included in the discussion. There is no study we are aware of that compared NaPIC and/or Cav1.1 currents in humans with myotonia congenita to controls.

4. Since the incidence of PPs correlates with firing frequency during myotonia, the authors propose that increases in intracellular calcium could modulate the activity of channels responsible for the PP (lines 179-180). However, it not clear why this could not simply reflect voltage dependent changes in channel activity produced during the myotonia. Alternatively, higher firing rates would produce greater K accumulation and TT depolarization, which could also contribute to the observed correlation between firing frequency and incidence of PPs. In order to support the authors' preferred suggestion for a role of calcium-dependent mechanisms, experiments to more directly test this idea are needed (e.g. effect BAPTA-AM loading on firing rate, membrane resistance and plateau duration).

We have re-written the manuscript to decrease the emphasis on intracellular Ca^2+^ and included discussion of other factors that correlate with firing rate. We hope in the future to perform imaging studies of intracellular Ca^2+^ during plateau potentials.

5. In Figure 6B, it would be helpful if the authors could also replot the untreated WT force data from Figure 1E as this would allow readers to more directly assess the relative degree of rescue by ranolazine treatment compared to control WT mice.

We have made the requested change to the figure (now Figure 8B).

6. Unfortunately, force measurements in non-conducting Cav1.1. mice were not performed. Nevertheless, given the observed effect of Cav1.1 conductance on both PP duration and slope of repolarization as shown in Table 2, the authors should at least discuss the potential importance of Cav1.1 conductance in the transient weakness in myotonia congenita in the second to last paragraph of the Discussion.

We have added this point to the indicated paragraph.

7. The summary figure presented in the Discussion section (Figure 7) should also include information regarding the proposed role of NaPIC and CaV1.1 conductances in PP incidence and duration, respectively, which is the main contribution of this study.

The summary figure has been modified as requested.

Reviewer #3 (Recommendations for the authors):1. Page 14, line 291, perhaps state "by hyperpolarizing the mean membrane potential". The interpretation of a "decreasing" membrane potential can be ambiguous. Decreasing toward 0 mV (less polarized) or decreasing meaning more negative value (hyperpolarized).

The rephrasing has been incorporated.

2. Page 20, line 441, "… a subset of NaV1.4 channels that lack fast inactivation". This gives the impression of two separate functional classes of NaV1.4 channels: those the fast inactivate (the overwhelming majority) and that do not fast inactivate (a small minority). Instead, Dr. Patlak (1986) proposed modal gating, whereby with very low frequency any NaV1.4 channel may enter the non-inactivating mode, and then subsequently convert back. This remains the prevailing view.

We have altered the text as requested.

Reference:

Hawash AA, Voss AA, Rich MM (2017) Inhibiting persistent inward sodium currents prevents myotonia. Ann Neurol 82:385-395.